# Early-Onset HTLV-1-Associated Myelopathy/Tropical Spastic Paraparesis

**DOI:** 10.3390/pathogens9060450

**Published:** 2020-06-07

**Authors:** Alvaro Schwalb, Valeria Pérez-Muto, Rodrigo Cachay, Martín Tipismana, Carolina Álvarez, Fernando Mejía, Elsa González-Lagos, Eduardo Gotuzzo

**Affiliations:** 1Instituto de Medicina Tropical ‘Alexander von Humboldt’, Universidad Peruana Cayetano Heredia, Lima 15102, Peru; rodrigo.cachay.f@upch.pe (R.C.); martin.tipismana.b@upch.pe (M.T.); carolina.alvarez@upch.pe (C.Á.); fernando.mejia.c@upch.pe (F.M.); elsa.gonzalez@upch.pe (E.G.-L.); eduardo.gotuzzo@upch.pe (E.G.); 2School of Medicine, Universidad Peruana Cayetano Heredia, Lima 15102, Peru; valeria.perez.m@upch.pe; 3Hospital Cayetano Heredia, Lima 15102, Peru

**Keywords:** HAM/TSP, myelopathy, paraparesis, chronic infection, inflammation

## Abstract

Background: Vertical transmission of HTLV-1 could lead to the early development of HTLV-1-associated myelopathy/tropical spastic paraparesis (HAM/TSP). This significantly affects quality of life and increases morbimortality. Objective: To describe the epidemiological and clinical characteristics of patients with early-onset HAM/TSP, defined as disease onset before 20 years of age. Methods: This is a retrospective study from an HTLV-1 clinical cohort between 1989 and 2019. We searched for medical records of patients with (1) diagnosis of HTLV-1 infection using two ELISA and/or one Western blot, (2) clinical diagnosis of HAM/TSP by neurological assessment, and (3) HAM/TSP symptom-onset before 20 years of age. Results: A total of 38 cases were identified in the cohort; 25 were female (66%). The median age of onset was 14 years old. 31 (82%) cases had HTLV-1 testing done among family members; 22 out of 25 tested mothers (88%) were HTLV-1 positive. Most patients (27/34) were breastfed for more than one year. Disease progression measured through EDSS and IPEC-1 showed an upward trend towards worsening spasticity with 18 patients (47%) eventually requiring mobility aids. Conclusions: Cases of early-onset HAM/TSP are not of rare occurrence, which translates into many more years of dependency, the use of mobility aids, and increased overall morbidity.

## 1. Introduction

Globally, approximately 5 to 10 million people are infected with the human T-cell lymphotropic virus type 1 (HTLV-1) [1]. In Peru, the HTLV-1 prevalence in candidate blood donors varies from 1.2 to 1.7% across regions, with higher figures for some Andean areas [2]. While the vast majority of people with HTLV-1 may remain asymptomatic for decades, approximately 5–10% will present adverse health outcomes, which are still being characterized [3].

HTLV-1-associated myelopathy/tropical spastic paraparesis (HAM/TSP) is the inflammatory disease mostly reported in association with this retroviral infection [4]. Approximately 0.25% to 3.8% of HTLV-1 carriers will develop HAM/TSP [5,6]. The predominant features of HAM/TSP are neurological; it evolves as a slow-progressing demyelinating disease of the central nervous system that manifests with gradual spastic paraparesis, neurogenic bladder, gait disturbances, and many mild sensory signs [7]. This condition predominantly affects female patients and is usually diagnosed at the age of 40 or 50, although, by the time of diagnosis, most patients had endured symptoms for over a decade [8,9].

Generally considered an adult manifestation of HTLV-1, an early infection could lead to the early development of HAM/TSP; in fact, publications are increasingly pointing to early juvenile HAM/TSP [10,11,12]. In HTLV-1-endemic areas, mother-to-child transmission, linked to extended breastfeeding, is quite important; the rate of seroconversion is 15.7% among children who have been breastfed for more than six months [13]. Additional factors for vertical transmission, apart from the duration of breastfeeding, include high proviral load (PVL) in the mother and, consequently, in breast milk [14]. Oral infection during breastfeeding occurs through unknown mechanisms that may be associated with high levels of PVL in breastmilk and a low immune response from the child [15]. Early acquisition of HTLV-1 might increase the odds of adverse health outcomes, and an increase in the cumulative risk of developing adult T-cell leukemia/lymphoma (ATLL) [16,17].

The development of HAM/TSP early in life may contribute to greater disease progression, significantly affecting the quality of life of the afflicted and increasing morbimortality due to related conditions [9]. In collaboration with Hospital Cayetano Heredia, the Instituto de Medicina Tropical Alexander von Humboldt (IMTAvH) in Lima has established clinical services for people with HTLV-1 for more than 30 years and manages a cohort of people with HTLV-1 infection. In this study, we describe the epidemiological characteristics of patients with early-onset HAM/TSP, which we define as disease onset before 20 years of age.

## 2. Results

A total of 38 cases were identified in the cohort; 25 were female (66%). Three cases were siblings. The main reason for inclusion into the HTLV-1 cohort was the presence of HAM/TSP symptomatology (84%), followed by having a family member with a positive HTLV-1 test (13%). One patient was included due to a diagnosis of infective dermatitis (ID). The median age of onset of HAM/TSP was 14 years old; though, it was lower (12) for the 15 patients with a history of ID. The earliest reported symptom-onset age was 6 years old. The median duration of symptoms prior to the HAM/TSP diagnosis was 6 years (Table 1).

Most patients (79%) had been breastfed for more than one year. Six patients reported a history of blood transfusion, of whom three had a mother who was HTLV-1 positive. By geographical region, 22 had mothers born in the highlands, predominantly from Apurimac, and the rest came from the coastal region, mostly from Lima (Figure 1).

HTLV-1 testing among family members was done in 31 patients (82%); 26 of them had at least one HTLV-1 positive family member. 22 out of 25 mothers (88%) who were tested had HTLV-1 infection and 10 were diagnosed with HAM/TSP. Out of 20 patients with a history of at least one family member with walking difficulties, in 14 this was attributable to HAM/TSP. At least 13 patients had multiple family members with positive HTLV-1 status including fathers, siblings, spouses, and children.

The median time of follow-up after the initial evaluation and HAM/TSP diagnosis was 4.5 years, although 11 patients were lost to follow-up after less than a year or immediately after the initial diagnosis of HAM/TSP. The median age of the cohort at last follow-up was 29.5 years (IQR 21.5–35.75). During follow-up, 29 patients were prescribed baclofen as standard of care. However, information on adherence is not clear; at least 9 patients self-reported taking medication irregularly or having stopped it altogether for months or years, and 11 of them have been lost at follow-up for more than 5 years. Additionally, 27 patients received other types of treatments including vitamin C (11), corticosteroids (9), analgesics (6), antiretrovirals (5), benzodiazepines (4), laxatives (4), ferrous sulfate (3), orphenadrine (3), oxybutynin (3), gabapentin (2), antidepressants, and tolterodine. At least 4 patients out of 27 were referred for physical therapy.

Dermatologic manifestations of HTLV-1 were reported by 25 patients (66%) and are listed in Table 2. The most common was ID, which was reported by 15 patients; in eight of them ID preceded HAM/TSP symptom onset. Ten cases had scabies, and seven patients reported having been diagnosed with both ID and scabies. There were ophthalmologic disturbances in 15 patients; three of them were diagnosed with corneal ulcers.

The Kurtzke Expanded Disability Status Scale (EDSS) was measured in 20 patients, nine of whom had one additional measurement during their follow-up. Additionally, the Instituto de Pesquisa Clínica Evandro Chagas (IPEC-1) scale was measured in 20 patients. Figure 2 plots the EDSS value progression for these nine cases by time since symptom onset. The final EDSS score median in 20 patients was 6.0, measured after a median of 9.5 years since symptom onset. Figure 3a,b plot the highest EDSS and IPEC-1 scores, respectively, since symptom onset with a regression line with a tendency of 0.08 EDSS and 0.09 IPEC-1 score increase per year.

A total of 18 patients (47%) required mobility aids during disease progression, most commonly walking canes (9/18) and wheelchairs (8/18). Bladder control disturbances were reported by 33 (87%) patients. Values of HTLV-1 proviral load (PVL) were only available for 12 patients with a median PVL of 3125.5 copies per 10^4^ PBMCs (IQR: 1658–4363, Range: 773–5125).

## 3. Discussion

HTLV-1 patients infected via vertical transmission can manifest HAM/TSP from a younger age. Such cases are not uncommon; one study found 27 cases of early-onset HAM/TSP reported in publications, and, recently, another study followed and described 25 juvenile patients with HAM/TSP [10,11]. The 38 patients identified in our study further contribute to the growing research on the subject.

For most young HTLV-1 carriers, vertical transmission occurred during infancy via breastfeeding. Gotuzzo et al. found a transmission rate of 19% among HTLV-1-infected mothers and their offspring [18]. In our study, 22 mothers tested positive for HTLV-1, which suggests that most of our early-onset HAM/TSP patients were infected vertically through breastfeeding. A meta-analysis found a pooled odds ratio of 3.48 (95%CI, 1.58–7.64, *p* = 0.002) of HTLV-1 transmission in breastfed infants compared to bottle-fed infants [19]; other studies report higher transmission rates when breastfeeding duration is greater than 12 months, as is the case for 27 of our patients [20]. Likewise, there is a lower seroconversion rate when the breastfeeding period is shorter (less than or equal to 6 months) [21]. Through the implementation of a prevention program in Nagasaki, which refrained HTLV-1-infected mothers from breastfeeding, there was a decrease in vertical transmission from 20.3% to 2.5% [13]. Although recommendations for suppression of breastfeeding can be implemented and directed to seropositive expectant mothers, national programs are currently unprepared to provide formula beyond the first 6 months of breastfeeding; therefore, such strategies may be justified in developed countries but not in countries with high socio-economic inequality like Peru [22]. This must be weighted against the fact that mother-to-child-transmission accounts for 30% of HAM/TSP cases [23].

Since there is a higher prevalence of HTLV-1 infection among people from certain regions and ethnic groups [24], we took into account the mother’s birthplace. Accordingly, 22/38 mothers were born in the highlands region of Peru and 7/38 were born in Lima (including Callao province). Studies from the last 20 years carried out in Peru reveal high seroprevalence of HTLV-1 not only in women born in the Andean regions, but also in those born in Lima [25,26]. This higher prevalence in Lima is due to the greater mestizo population in the country’s capital after considerable migration from the Andean regions over the last decades [27].

Vertical transmission paves the way for familial aggregation of HTLV-1. A systematic review identified 270 families in which more than one family member had an HTLV-1-associated disease; of these families, 102 had several family members with HAM/TSP [28]. HTLV-1 testing was done among family members of 31 patients; 26 (84%) had at least one seropositive family member and 13 (42%) had multiple seropositive relatives. As mentioned before, in this study, 22 mothers were HTLV-1 positive and 10 had also been diagnosed with HAM/TSP. Three patients included in this study belong to the same family with a high burden of HAM/TSP [29]. Previous cohorts that have identified 5.1% to 22% HAM/TSP prevalence among family members show significant differences in the age of onset and progression speed between cases of HAM/TSP with and without familial history [30,31]. Familial clustering of HAM/TSP may be due to the genetic characteristics of HTLV-1 carriers [32]. The potential for clustering reinforces the need for measures that prevent vertical transmission.

Our cohort used the EDSS and IPEC-1 to evaluate disease severity and, in some cases, progression. Among those patients whose scores were measured on two separate occasions, there is a clear increasing trend confirming that disease severity worsens through the years (Figure 2 and Figure 3a,b). Another study evaluating disease progression in early-onset HAM/TSP patients demonstrated a similar upward trend, with multiple measurements [10]. Studies on adult-onset HAM/TSP patients described EDSS score medians of 6.0 after 8–11 years [33,34]. On the other hand, another study reports a slow disease progression in patients with HAM/TSP onset before the age of 15 when compared to older patients [8]. Although our patients present an upward trend in disease progression, early-onset HAM/TSP progression does not seem to be faster than the reported progression in adult-onset HAM/TSP patients. Regardless of the rate of progression or severity, there is no doubt that the disabling consequences of the disease affect the patients’ quality of life from an early age [11].

Studies on adult HAM/TSP patients described that 46.1–50% of them needed a walking aid after a decade of disease progression; wheelchairs were frequently required [31]. Similarly, in our study, 18 (47.4%) patients with early-onset HAM/TSP began using walking aids during follow-up, 8 of whom reported the use of wheelchairs. The need for mobility aids correlates to more incapacitating disease progression and greater dependency. It also predisposes the morbidity associated with prostration, such as muscular atrophy and infections, and has a very negative impact on mental health and quality of life [35]. These side effects are further complicated by urinary disturbances (usually vesical hyperactivity), which have been reported in 84.1% of HAM/TSP cases [36]. Although no urodynamic studies were performed among our participants, 86.9% reported difficulties with bladder control. These symptoms greatly impair quality of life for patients experiencing them [36]. The fact that our patients started developing urinary disturbances at an early age implies that they spent most of their adult life dealing with related comorbidities and increased dependency.

Dermatologic manifestations are commonly diagnosed in the HTLV-1-infected population, but such symptoms may be as high as 88% in HAM/TSP patients [37]. According to Okajima et al., the most frequently diagnosed skin conditions in HTLV-1 adult patients are xerosis (49.2%), infestations including scabies (31%), and seborrheic dermatitis (28%) [37]. Other, less frequent skin disorders include herpes zoster, pityriasis, warts, and contact dermatitis. We reported 23 patients (61%) with at least one dermatologic manifestation.

Of the many dermatological manifestations that have been associated with HTLV-1, the most frequently observed, especially in vertically-infected patients, is ID. This is commonly associated with HAM/TSP and ATLL, and some studies suggest that ID could serve as a disease marker [38,39]. Among a cohort of 37 young HTLV-1 infected patients with ID, 54% developed early-onset HAM/TSP [39]. Similarly, 17 cases of early HAM/TSP were diagnosed among patients with a history of ID [11]. We found that ID was the most frequently reported dermatological manifestation and that it preceded HAM/TSP in half of the cases. Furthermore, the median age of symptom onset has been found to be significantly younger in those with a history of ID (9 vs. 16 years old; *p* = 0.045) [10]; this difference existed among our patients but it was not statistically significant (12 vs. 16 years old; *p* = 0.159). Given these results, our study further supports the evidence of an association between ID and HAM/TSP development.

HTLV-1 PVL has been considered a strong predictor of HAM/TSP development [40]. One study found a significant association between PVL and HAM/TSP while adjusting for other immune markers, and another study found an increase in PVL when clinical worsening was recorded in patients [41,42]. Higher PVLs have also been found in asymptomatic patients infected via breast milk compared to those infected through sexual contact, further suggesting that vertical transmission could be a greater risk factor for the development of HTLV-1-associated diseases [43]. Additionally, PVL concentrations are significantly higher in cerebrospinal fluid compared to those in peripheral blood mononuclear cells (PBMCs) from the same patients, indicating that high PVL concentration plays a key role in CNS affliction and HAM/TSP pathogenesis [44]. Lastly, higher PVLs are also found in patients with ID when compared to asymptomatic carriers [45]. Only 12 of our patients had PVL values available, in which the median was 3125.5 copies per 10^4^ PBMCs; of note, these patients had a history of ID. Although they were taken at different times during follow-up, this median value is indicative of HTLV-1-associated inflammatory manifestations [41,45].

Despite small studies and a few randomized trials evaluating treatment options aimed at improving clinical manifestations and reducing PVL in HAM/TSP patients, to date, no standardized treatment has been established [6]. In most cases, patients are given an array of different medications, which could include antiretrovirals, corticosteroids, immunosuppressive drugs, and monoclonal antibodies, although there is not sufficient evidence confirming long-term improvement [6]. Among our patients, the most frequently used drug was baclofen, which was prescribed for 29 (76%) patients. Baclofen has been used in patients with multiple sclerosis, improving spasticity and reducing the frequency of spasms and clonus [46]. Most of our patients (71%) also received other drugs, including vitamin C (29%), corticosteroids (24%), analgesics (16%), and even antiretrovirals (13%). Vitamin C was studied in an open-label trial (n = 7) with intermittent high-doses, decreasing disability score after 9.7 months [47]. Likewise, corticosteroids have been widely used for the management of HAM/TSP, although most positive effects have been limited and only observed in a few case-series [46]. Although our patients received different treatments, these vary in duration, adherence, doses and combinations, which poses an obstacle in the evaluation of their benefits to patient manifestations.

Our study faced several limitations, including the heterogeneity of the available information, recall bias, and unstandardized patient follow-up. As this is a retrospective study, complete data could not be obtained from the patients’ charts and the cohort database, particularly among patients who only completed the initial visit or had fewer follow-ups. For instance, less than half of all patients had a PVL value and not every one of them had HTLV-1 testing done among family members, which could have provided further data on familial clustering. Furthermore, recall bias may have affected the patient’s responses to onset date, symptoms, and prior dermatologic manifestations that could have occurred early in life. Also, the patient follow-up and treatment were not standardized and, therefore, it was not possible to evaluate treatment impact on progression. Different scales for disease progression and varying time points for PVL measurements at varying complicated data harmonization. Additionally, early EDSS scores were often unavailable due to minimal symptoms and delayed diagnosis.

Cases of early-onset HAM/TSP are not of rare occurrence; the disease is one of the most common manifestations of vertically-transmitted HTLV-1. HAM/TSP not only comprises gait disturbances but also affects urinary functions and has a paramount impact on mental health and overall quality of life. We found that earlier onset of HAM/TSP symptoms translated into many more years of dependency, the use of mobility aids, and increased overall morbidity. The patients from our study highlight the need for antenatal screening in pregnant women in endemic settings to prevent HTLV-1 transmission and its complications. Furthermore, this work reaffirms the importance of family studies and a close clinical follow-up of the offspring of HTLV-1 positive women. Future directions for the care of early-onset HAM/TSP should include meticulous and extensive follow-up and implementation of physiotherapeutic treatments aimed at delaying disease progression.

## 4. Materials and Methods

This is a retrospective study of HTLV-1 infected patients from the IMTAvH clinical cohort between 1989 and September 2019. Since its inception, the cohort has offered routine evaluation of people with confirmed and suspected HTLV-1 infection. Evaluations include a standardized cohort-inclusion form performed by trained healthcare workers, a thorough medical evaluation from various specialties, and HTLV-1 testing for family members. Patients with HTLV-1 infection with suspected HAM/TSP are examined by an infectious disease specialist and by a neurologist. In this study, HAM/TSP clinical diagnosis was made based mainly on the presence of chronic spastic paraparesis, weakness of the lower limbs, and/or difficulty walking, running, dancing, or standing up. Other signs and symptoms, such as bladder disturbances, low lumbar pain, impaired vibration sense, and hyperreflexia in the lower limbs (clonus and Babinski’s sign) and the upper limbs (Hoffmann’s and Trommer signs), also supported HAM/TSP diagnosis [48]. HTLV-1 antibodies or PVLs in cerebrospinal fluid are not routinely performed to confirm HAM/TSP diagnosis.

The cohort database was searched for medical records of patients with (1) diagnosis of HTLV-1 infection using two enzyme-linked immunosorbent assays and/or one Western blot, (2) clinical diagnosis of HAM/TSP by the aforementioned neurological assessment, and (3) HAM/TSP symptom-onset before 20 years of age.

The data extracted from medical records and cohort patient questionnaires included clinical variables including sex, age at disease onset, duration of disease (considered from the beginning of aforementioned motor symptoms to diagnosis), duration of breastfeeding, PVL, and past blood transfusions. The presence of other diseases such as ID, scabies, and other dermatologic manifestations was also noted. Data concerning the patient’s family was also considered: the presence of a family member with HTLV-1 diagnosis or testing and the mother’s birthplace were recorded. Disease progression was measured with the Kurtzke EDSS, originally designed to assess motor disability in multiple sclerosis patients, and the IPEC-1 scale [49,50].

Patient information was registered in a Microsoft Excel database and frequencies and percentages were calculated using Stata SE 15 (StataCorp., College Station, TX, USA). Statistical analysis included the use of the Mann-Whitney U test. The study was approved by the Institutional Ethics Committee of UPCH (SIDISI: 103844) before the start of data collection.

## Figures and Tables

**Figure 1 pathogens-09-00450-f001:**
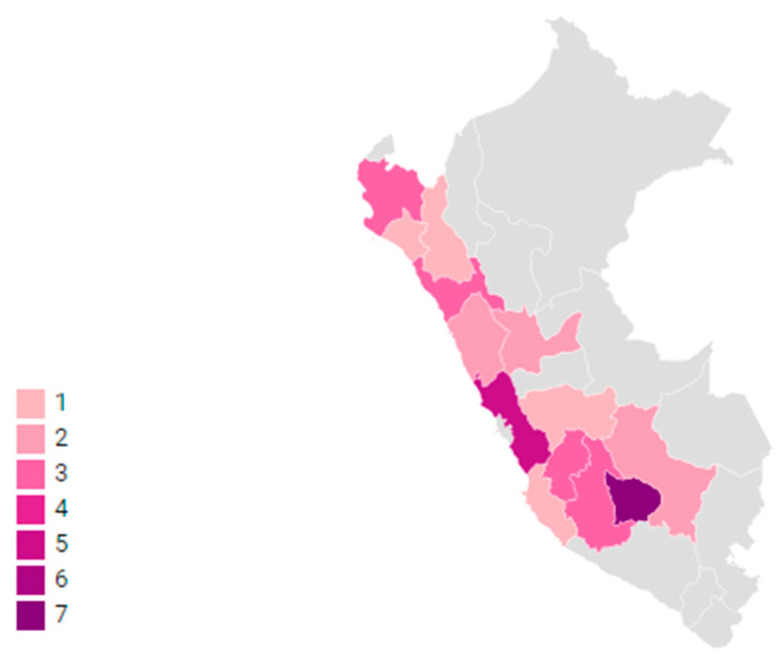
Birthplace of HAM/TSP patient’s mothers by regions in Peru. Color gradients represent the birthplace of the patients’ mothers per region in Peru. Map made with Datawrapper (https://app.datawrapper.de/map/).

**Figure 2 pathogens-09-00450-f002:**
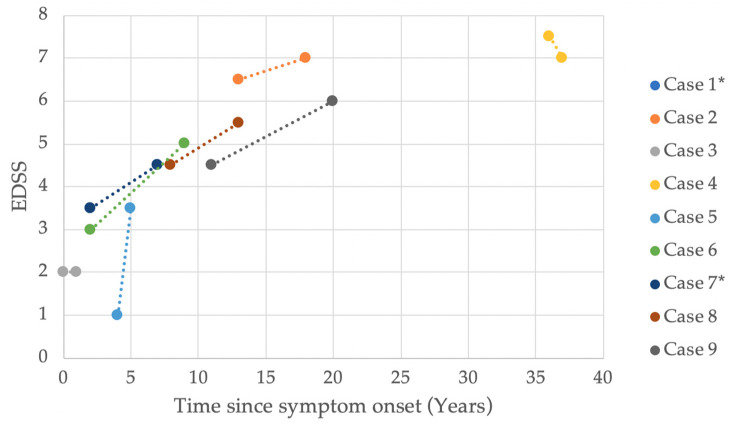
EDSS value progression by time since symptom onset in patients with early-onset HAM/TSP. EDSS: Kurtzke Expanded Disability Status Scale (EDSS). HAM/TSP: HTLV-1-associated myelopathy/tropical spastic paraparesis. * Case 1 and Case 7 experienced the same progression and overlap in the graph.

**Figure 3 pathogens-09-00450-f003:**
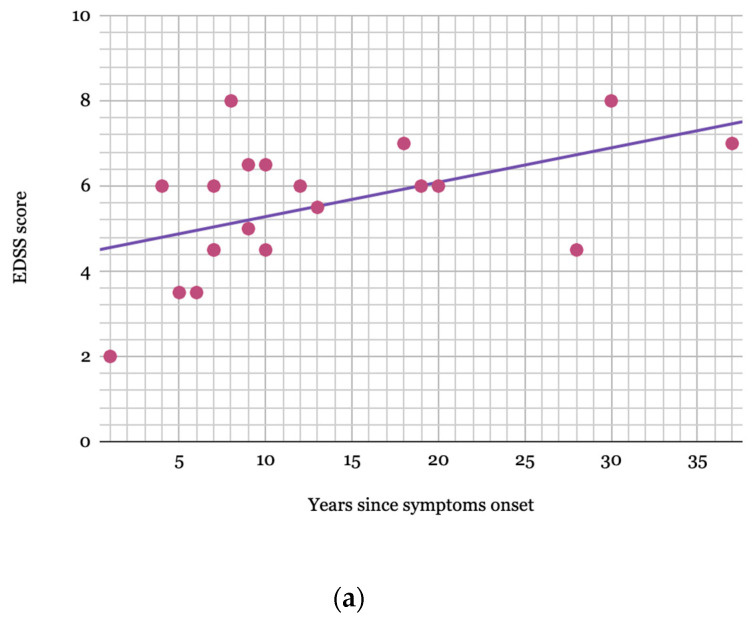
(**a**) Higher EDSS value since symptom onset in patients with early-onset HAM/TSP. EDSS: Kurtzke Expanded Disability Status Scale (EDSS). HAM/TSP: HTLV-1-associated myelopathy/tropical spastic paraparesis. Only 19 patients appear to be plotted as two presented the same EDSS score recorded in the same year since symptom onset. (**b**) Higher IPEC-1 value since symptom onset in patients with early-onset HAM/TSP. IPEC-1: Instituto de Pesquisa Clínica Evandro Chagas scale. HAM/TSP: HTLV-1-associated myelopathy/tropical spastic paraparesis.

**Table 1 pathogens-09-00450-t001:** Demographic and clinical characteristics of early-onset HAM/TSP patients.

Demographic/Clinical Characteristics (Total = 38)	n (%) ^1^
Female	25 (66)
HTLV-1 diagnosis made due to HAM/TSP symptoms	32 (84)
Duration of breastfeeding >1 year (n = 34)	27 (79)
History of blood transfusion	6 (16)
Patients with family members tested for HTLV-1	31 (82)
*Mothers with confirmed HTLV-1 diagnosis*	22 (71)
*Patients with other family members with HTLV-1 infection*	17 (55)
A family member with walking difficulties attributable to HAM/TSP	14 (37)
Mothers diagnosed with HAM/TSP	10 (26)
Age of onset of HAM/TSP symptoms, years, median (IQR)	14 (7)
*Female patients*	14 (5)
*Male patients*	16 (6)
Duration of symptoms prior to diagnosis, years, median (IQR)	6 (7)
Follow-up duration, years, median (IQR)	4.5 (13)

^1^ Values are n (%) unless noted otherwise. HTLV-1: human T-cell lymphotropic virus Type 1. HAM/TSP: HTLV-1-associated myelopathy/tropical spastic paraparesis. IQR: Interquartile range.

**Table 2 pathogens-09-00450-t002:** Dermatologic manifestations of patients with early-onset HAM/TSP.

Dermatologic Manifestations (Total = 38)	n (%) ^1^
Infective dermatitis	15 (40)
Scabies	10 (26)
Onychomycosis	14 (37)
Warts	9 (24)
Seborrheic dermatitis	3 (8)
Herpes zoster	2 (5)
Other ^2^	5 (13)

^1^ Values are n (%). ^2^ Allergic dermatitis, psoriasis, alopecia areata, pityriasis, and pitted keratolysis.

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
