# Peer review of "Early-Onset HTLV-1-Associated Myelopathy/Tropical Spastic Paraparesis"

_pathogens, 2020, doi:10.3390/pathogens9060450_

Round 1
Reviewer 1 Report
Schwalb et al. in their manuscript entitled “Early-Onset HTLV-1 Associated Myelopathy/Tropical Spastic Paraparesis” provide a retrospective study from an HTLV-1 clinical cohort. The goal of this research was to describe the epidemiological and clinical characteristics of patients with early-onset HAM/TSP. The authors found a majority of patients in their cohort were female (66%) and media age of HAM/TSP onset was 14 years old. Interestingly, disease progression in this study was found to have an upward trend, but not significantly faster than progression in adult-onset HAM/TSP. This study provides additional evidence that early-onset HAM/TSP is not a rare occurrence, as previously thought. The manuscript is well written and the data support the conclusions.
Only one minor question: Is there any evidence to suggest early-onset HAM/TSP patients have a greater likelihood of vertical transmission to future offspring or a greater likelihood of ATL development?
Author Response
Dear Reviewer,
We appreciate your comments and the appraisal of our manuscript. Regarding your minor question, the unstandardized patient follow-up in our study does not allow us to reach any evidence regarding the greater likelihood of vertical transmission to future offspring or a greater likelihood of ATL development. However, other studies do provide evidence of this.
One study presented modeling of the risk of ATL and found that childhood-acquired HTLV-1-seropositivity was an important factor in the development of ATL. This study also finds that among people infected before age 20, the cumulative lifetime risk of developing ATL is 4.0% and 4.3% for males and females, respectively. (Murphy, Int J Cancer 1989). We have included this in the manuscript (Introduction, Page 2, Line 53-54)
Furthermore, there is currently no evidence that might suggest that early-onset HAM/TSP patients have a greater likelihood of vertical transmission to their offspring. Although differences in PVL values, and therefore the risk of vertical transmission, exist between HAM/TSP patients and asymptomatic carriers, at present to our knowledge there aren't any studies available comparing early and late-onset HAM/TSP.
We look forward to hearing from you at your earliest convenience.
Yours sincerely,
Alvaro Schwalb
Reviewer 2 Report
This manuscript entitled “Early-onset HTLV-1 associated myelopathy/tropical spastic paraparesis” examines the clinical characteristics as well as dermatologic and neurologic manifestations of juvenile patients with HAM/TSP in Peru. Various symptoms are severe even if HAM/TSP develops while the patient is young. It is important to present the details of various clinical manifestations associated with HAM/TSP before 20 years of age. However, several points should be addressed, including the following:
- Early-onset HAM/TSP are not rare occurrences. How often did early-onset HAM/TSP occur in all HAM/TSP patients or HTLV-1 infected individuals in this study?
- Although the EDSS was measured in 20 patients, there are 19 plots in Figure 3a.
- Figure legends should be added.
- Page 4 line 10. “Figure 3a” should be “Figure 3b.”
Author Response
Dear Reviewer,
Thank you for your comments and the appraisal of our manuscript. The responses to your comments are as followed:
1. This statement has been made according to the tendency to consider HAM/TSP an adult-onset disease. Other similar studies on early-onset HAM/TSP also address this issue (See Varandas CID 2018 and Oliveira J Trop Pediatr 2018). We did not feel confident with our cohort's denominator estimates to make a numeric claim.
2. Two patients overlap on EDSS value and years since symptom onset (EDSS: 4.5, 7 years since symptom onset), hence, only 19 patients appear to be plotted. We have included this is the Legend of Figure 3a (Page 4)
3. We have added legends to all figures.
4. This has been corrected.
We hope to hear from you at your earliest convenience.
Yours sincerely,
Alvaro Schwalb
Reviewer 3 Report
The manuscript provides a retrospective description of 38 patients with the Lima cohort with onset of symptoms of HAM/TSP before age 20. Compared to other publications in the field this is a large number of early onset cases and provides important new perspectives.
The manuscript might be further improved:
1) The primary conclusion is that cases of early onset HAM are not rare. Please include the size of the cohort (and the number of patients with HAM) from which these patients were identified. What % have first symptoms prior to 20 years age. Would the authors like to comment on the relative contribution of MTCT of HTLV-1 to the total burden of disease caused by HAM. This could be both the % of patients but also the attributable disease-years.
2) Please state the units (per 104 PBMC) of the proviral load in the results. The median of 3125/104 appears high - perhaps due to the cases of ID? How does this compare with median proviral of all other patients with HAM in this cohort?
3) What is the median age of the cohort at last FU? Did any patients develop ATL?
4) The second conclusion is that early onset HAM results in years of dependency. 47% of the cohort required walking aids at the time of analysis. What was the median interval from onset to first use of walking aid and to wheelchair dependency?
5) Are ambulant patients more likely to continue in FU than those with mobility issues?
6) In a high proportion of patients with early onset HAM, where tested, mothers were HTLV-1 seropositive. Considering the three where this was not the case where they older at presentation and thus potentially might have acquired HTLV infection through sexual intercourse? Is wet nursing a possibility.
7) What was the earliest age of first gait related symptoms in this cohort?
Author Response
Dear Reviewer,
Thank you for your comments and the appraisal of our manuscript. We currently have limited access to the cohort's database and patient records due to our country's lockdown. The responses to your comments are as followed:
1. This statement has been made according to the tendency to consider HAM/TSP an adult-onset disease. Other similar studies on early-onset HAM/TSP also address this issue (See Varandas CID 2018 and Oliveira J Trop Pediatr 2018). The database search was focused on the inclusion criteria for the patients included in the study, and we do not feel confident with our cohort's denominator estimates to make a numeric claim, as this value can be over or underestimated as the whole database has not been curated. Regarding the contribution of MTCT to the burden of HAM/TSP, we have added a sentence addressing an estimate. (Page 5, Line 158-160)
2. We have stated the units for the median PBMCs in the results, and have provided better information on the IQR. (Page 5, Line 137). We have mentioned that the PVL provided all come from patients with a history of ID, as you suspected in your comments (Page 7, Line 230-231). As mentioned above, we don't have full access to the database, and therefore, don't have the PVL values.
3. None of the patients presented in our study developed ATL. We included the median age of the cohort at last follow-up in the results sections, where data on follow-up is presented (Page 3, Line 92-93).
4. We were unable to gather the dates from when walking aids start being reported. This data might be very interesting to obtain and compare to the use of walking aids in cases of adult-onset in a future study.
5. This point raised is similar to the one above. There is no registry on why patients might choose not to attend FU visits. From our experience, the cohort's nurses do a tremendous job to help patients with their check-ups however, this might be an interesting survey to conduct while gathering information of the impact of walking aids.
6. In our study, other ways of acquiring HTLV-1 besides breastfeeding were possible. Wet nursing is not a common practice. Two of these patients developed reported symptom onset at 19 and 16 years of age, so sexual intercourse is a possibility; the latter also reported blood transfusion. The other patient is the legal case of a child, with extensive testing among family members, no wet nursing, and no reported blood transfusions. No source was determined but was attributed to a minor dental procedure. We don't feel that these explanations contribute further to the study as they are speculative.
7. The earliest reported symptom-onset age in our cohort is 6 years old (Page 2, Line 68-69); however, it is not clear if this was a gait-related symptom.
We would like to thank you for providing us with a future research idea studying patients requiring walking aids from our cohort. We hope to hear from you at your earliest convenience and apologize again for our limited access to the resources that might have further replied your review.
Yours sincerely,
Alvaro Schwalb